# A Bibenzyl from *Dendrobium pachyglossum* Exhibits Potent Anti-Cancer Activity Against Glioblastoma Multiforme

**DOI:** 10.3390/antiox14101212

**Published:** 2025-10-07

**Authors:** Hnin Mon Aung, Onsurang Wattanathamsan, Kittipong Sanookpan, Aphinan Hongprasit, Chawanphat Muangnoi, Rianthong Phumsuay, Thanawan Rojpitikul, Boonchoo Sritularak, Tankun Bunlue, Naphat Chantaravisoot, Claudia R. Oliva, Corinne E. Griguer, Visarut Buranasudja

**Affiliations:** 1Pharmaceutical Sciences and Technology Program, Faculty of Pharmaceutical Sciences, Chulalongkorn University, Bangkok 10330, Thailand; 6671006433@student.chula.ac.th (H.M.A.); 6573002133@student.chula.ac.th (T.R.); 2Department of Pharmacology and Physiology, Faculty of Pharmaceutical Sciences, Chulalongkorn University, Bangkok 10330, Thailand; onsurang.w@chula.ac.th (O.W.); sanookpan.k@gmail.com (K.S.); aphinan.h@pharm.chula.ac.th (A.H.); 3Biological Science and Animal Model Unit, Institute of Nutrition, Mahidol University, Nakhon Pathom 73170, Thailand; chawanphat.mua@mahidol.ac.th (C.M.); rianthong.phu@mahidol.ac.th (R.P.); 4Department of Pharmacognosy and Pharmaceutical Botany, Faculty of Pharmaceutical Sciences, Chulalongkorn University, Bangkok 10330, Thailand; boonchoo.sr@chula.ac.th; 5Center of Excellence in Natural Products for Ageing and Chronic Diseases, Faculty of Pharmaceutical Sciences, Chulalongkorn University, Bangkok 10330, Thailand; 6Department of Biochemistry, Faculty of Medicine, Chulalongkorn University, Bangkok 10330, Thailand; 6771008030@student.chula.ac.th (T.B.); naphat.c@chula.ac.th (N.C.); 7Center of Excellence in Systems Microbiology, Faculty of Medicine, Chulalongkorn University, Bangkok 10330, Thailand; 8Center of Excellence in Systems Biology, Faculty of Medicine, Chulalongkorn University, Bangkok 10330, Thailand; 9Free Radical & Radiation Biology Program, Department of Radiation Oncology, University of Iowa, Iowa City, IA 52242, USA; claudia-oliva@uiowa.edu (C.R.O.); cgriguer@healthcare.uiowa.edu (C.E.G.)

**Keywords:** anti-cancer, phytochemical, bibenzyl, *Dendrobium*, glioblastoma

## Abstract

Glioblastoma multiforme (GBM) is an aggressive brain tumor with limited treatment options and a poor prognosis. Natural phytochemicals from *Dendrobium* species, particularly bibenzyl derivatives, possess diverse pharmacological activities, yet their potential against GBM remains largely unexplored. Here, we investigated the anticancer activity of 4,5,4′-trihydroxy-3,3′-dimethoxybibenzyl (TDB), a potent antioxidant bibenzyl derivative isolated from *Dendrobium pachyglossum*. In U87MG cells, TDB reduced viability in a dose- and time-dependent manner, suppressed clonogenic growth, induced apoptosis *via* Bax upregulation and Bcl-xL/Mcl-1 downregulation, and inhibited both mTORC1 and mTORC2 signaling. TDB also impaired cell migration and downregulated epithelial–mesenchymal transition (EMT)-associated proteins. Notably, TDB enhanced the cytotoxicity of temozolomide (TMZ), the current standard of care for GBM. These TMZ-sensitizing properties were further confirmed in patient-derived xenograft (PDX) Jx22 cells. To assess its potential for central nervous system delivery, blood–brain barrier (BBB) permeability was predicted using four independent *in silico* platforms—ADMETlab 3.0, LogBB_Pred, LightBBB, and BBB Predictor (Tree2C)—all of which consistently classified TDB as BBB-permeable. This predicted CNS accessibility, together with its potent anticancer profile, underscores TDB’s translational promise. Collectively, our findings identify TDB as a plant-derived antioxidant with multifaceted anti-GBM activity and favorable BBB penetration potential, warranting further *in vivo* validation and preclinical development as a novel therapeutic candidate for GBM.

## 1. Introduction

Glioblastoma multiforme (GBM) is the most aggressive and frequently diagnosed malignant brain tumor in adults, classified as Grade IV by the World Health Organization. Despite advances in neurosurgery, radiotherapy, and chemotherapy, especially the use of temozolomide as a first-line agent, patient outcomes remain poor, with a median survival of just over a year. GBM’s invasive growth into surrounding brain tissue and its resistance to standard treatments contribute to inevitable recurrence and treatment failure [1,2,3]. These challenges emphasize the critical need for novel therapeutic strategies that can overcome resistance and improve long-term survival.

Plant-derived natural products continue to offer promising avenues for drug discovery due to their chemical diversity and biological activity. In fact, a substantial proportion of modern anticancer drugs have been either directly derived from phytochemicals or developed based on their structural frameworks [4]. Among them, species from the *Dendrobium* genus—orchids widely distributed in Asia and Oceania—have long been recognized in traditional medicine systems for their therapeutic potential. Recent studies have begun to uncover the anti-cancer properties of certain *Dendrobium*-derived compounds, including their ability to modulate cell proliferation, induce apoptosis, and interfere with tumor-related signaling pathways across various cancer models [5,6,7,8,9,10,11]. However, their activity against brain tumors such as GBM remains underexplored, representing an important gap in the field and a potential direction for innovative therapeutic development.

Among the bioactive metabolites isolated from *Dendrobium* species, bibenzyls have emerged as particularly promising candidates due to their structural uniqueness and diverse pharmacological activities. These phenolic compounds exhibit a broad spectrum of therapeutic properties, including antioxidant [12,13], anti-inflammatory [13], anti-bacterial [10], anti-diabetic [14], and, notably, anti-tumor effects [5,6,7,8,9,10,11]. In various cancer models, bibenzyls have been shown to suppress tumor cell proliferation, promote apoptosis, and inhibit migration, invasion, and metastasis [5,6,7,8,9,10,11]. One bibenzyl compound of particular interest is 4,5,4′-trihydroxy-3,3′-dimethoxybibenzyl (TDB), a structurally defined molecule identified in *Dendrobium* species and recognized for its potent bioactivity in preclinical models [6,7,11,15]. Despite its promising pharmacological profile, the anticancer potential of TDB in the context of GBM remains unexplored. This study aims to address this gap by evaluating the effects of TDB on GBM cell growth, survival, and migratory behavior, as well as its ability to potentiate the efficacy of standard chemotherapy.

In this work, we employed U87MG cells as a well-established *in vitro* GBM model to assess TDB’s anticancer properties [16]. Cytotoxicity was quantified using MTT assays, while clonogenic survival assays evaluated its long-term effects on tumor cell proliferation. The mode of cell death was characterized *via* flow cytometry and immunoblotting, with a focus on key apoptotic regulators. To explore potential molecular mechanisms, we assessed the impact of TDB on mTOR signaling, a central pathway implicated in GBM progression and therapeutic resistance [17]. Anti-migratory activity was examined through scratch-wound assays and analysis of epithelial–mesenchymal transition (EMT) markers. We also tested whether TDB could enhance the efficacy of temozolomide (TMZ), the current first-line chemotherapeutic for GBM. The anticancer effects of TDB were further examined in Jx22 patient-derived xenograft (PDX) cells, providing preliminary validation in a clinically relevant context. To evaluate its potential for central nervous system delivery, blood–brain barrier (BBB) permeability was assessed using four independent *in silico* prediction platforms—ADMETlab 3.0, LogBB_Pred, LightBBB, and BBB Predictor (Tree2C). Collectively, our findings identify TDB as a plant-derived compound with multifaceted anti-GBM activity and favorable CNS-access properties, supporting its potential as a promising therapeutic candidate.

## 2. Materials and Methods

### 2.1. Chemicals

Comprehensive information on the chemicals and reagents utilized in this study, together with manufacturer details and catalog numbers, is presented in Appendix A.

### 2.2. Isolation of 4,5,4′-Trihydroxy-3,3′-dimethoxybibenzyl (TDB)

4,5,4′-Trihydroxy-3,3′-dimethoxybibenzyl (TDB) was purified from the dried whole plants of *D. pachyglossum* as previously described [12]. Briefly, the plant material (2.7 kg) was macerated with methanol (MeOH), yielding a MeOH extract (220 g). The methanolic extract was mixed with water and subjected to partitioning with ethyl acetate (EtOAc) and *n*-butanol (*n*-BuOH) to give an aqueous extract (55 g), an EtOAc extract (110 g) and an *n*-BuOH extract (35 g). The ethyl acetate extract was fractionated by vacuum liquid chromatography on silica gel (EtOAc-hexane, gradient), yielding six fractions (A–F). Fraction E (11 g) was further separated by silica gel column chromatography (MeOH-CH_2_Cl_2_, gradient), resulting in four fractions (E1–E4). Fraction E2 (980 mg) was isolated by Sephadex LH-20 (acetone) and then subjected to column chromatography (silica gel, MeOH-CH_2_Cl_2_, gradient) to yield a brown amorphous solid of TDB (263 mg). The structure and purity (>98%) of TDB were determined by NMR analysis using a Bruker Avance DPX-300 FT-NMR spectrometer (Rheinstetten, Germany). Mass spectra were carried through a Bruker micro TOF mass spectrometer (ESI-MS) (Manchester, UK) (Figure 1); C_16_H_18_O_5_; HR-ESI–MS [M + Na]^+^ at *m/z* 313.1049 (calcd. for 313.1051, C_16_H_18_O_5_Na); ^1^H-NMR (300 MHz, acetone-*d*_6_) δ: 2.71 (2H, *m*, H_2_-α), 2.77 (2H, *m*, H_2_-α′), 3.75 (3H, *s*, MeO-3), 3.78 (3H, *s*, MeO-3′), 6.34 (1H, *d*, *J* = 2.0 Hz, H-2), 6.36 (1H, *d*, *J* = 2.0 Hz, H-6), 6.63 (1H, *dd*, *J* = 8.1, 2.0 Hz, H-6′), 6.71 (1H, *d*, *J* = 2.0 Hz, H-2′) and 6.78 (1H, *d*, *J* = 8.1 Hz, H-5′); ^13^C-NMR (75 MHz, acetone-*d*_6_) δ: 38.4 (C-α′), 38.8 (C-α), 56.2 (MeO-3′), 56.3 (MeO-3), 104.6 (C-2), 109.7 (C-6), 112.9 (C-2′), 115.5 (C-5′), 121.6 (C-6′), 132.7 (C-1), 133.8 (C-4), 134.2 (C-1′), 145.2 (C-4′), 146.0 (C-5), 148.0 (C-3′), 148.7 (C-3).

### 2.3. Ferric Reducing Antioxidant Power (FRAP) Assay

The FRAP assay is a widely used method for assessing the antioxidant capacity of a sample, based on its ability to reduce ferric (Fe^3+^) to ferrous (Fe^2+^) ions in the presence of the colorimetric reagent 2,4,6-tris(2-pyridyl)-s-triazine (TPTZ). Upon reduction, the Fe^2+^–TPTZ complex produces a blue color that can be quantitatively measured by spectrophotometry, with higher absorbance indicating greater reducing power [18].

For this study, the FRAP reagent was freshly prepared by combining 10 mL of 300 mM acetate buffer (pH 3.6) with ultrapure water to a final volume of 1 L. Separately, 1 mL of 10 mM TPTZ solution (dissolved in 40 mM HCl) was mixed with 1 mL of 20 mM ferric chloride hexahydrate solution in ultrapure water. These solutions were combined to yield the working FRAP reagent. In a 96-well plate, 20 μL of the sample was added to 150 μL of the FRAP reagent. The plate was incubated for 10 min at room temperature to allow complete color development. Absorbance was then measured at 600 nm using a CLARIOStar microplate reader (BMG Labtech, Ortenberg, Germany). Antioxidant capacity was quantified from a Trolox standard curve, and results were expressed as micromoles of Trolox equivalents per milligram of sample.

### 2.4. DPPH Free Radical Scavenging Assay

The 2,2-diphenyl-1-picrylhydrazyl (DPPH) assay is a widely used method for evaluating the antioxidant capacity of a sample, based on its ability to donate hydrogen atoms or electrons to neutralize DPPH free radicals. The stable violet DPPH radical exhibits a strong absorbance at 520 nm, which decreases upon reduction to its yellow-colored form, allowing quantification of radical scavenging activity [18].

For this study, the assay was performed with slight modifications from an established protocol [12]. Briefly, 20 µL of the test sample, dissolved in DMSO, was mixed with 100 µL of DPPH solution (150 µM) in a 96-well plate. The mixture was incubated at room temperature for 30 min in the dark to prevent light-induced degradation of DPPH. Absorbance was measured at 520 nm using a CLARIOStar microplate reader. A Trolox standard curve was used for calibration, and results were expressed as micromoles of Trolox equivalents per milligram of sample.

### 2.5. Cell Culture

The human glioma cell lines U87MG and H4 were purchased from the American Type Culture Collection (ATCC, Manassas, VA, USA). The patient-derived xenograft (PDX) GBM cell line Jx22 was generously provided by Dr. Jann Sarkaria (Mayo Clinic, Rochester, MN, USA [19]). U87MG and H4 cells were grown in Dulbecco’s Modified Eagle Medium (DMEM) supplemented with 10% fetal bovine serum (FBS), 100 U/mL penicillin, and 100 µg/mL streptomycin. Jx22 cells were maintained in a 1:1 mixture of DMEM and Ham’s F-12 (DMEM/F12) containing the same concentrations of FBS and antibiotics. All cultures were incubated at 37 °C in a humidified atmosphere of 95% air and 5% CO_2_. Mycoplasma contamination was routinely monitored using the Universal Mycoplasma Detection Kit (ATCC, Manassas, VA, USA).

### 2.6. Treatment with TDB or Temozolomide

Stock solutions of TDB and temozolomide were prepared in DMSO and stored at −20 °C until use. For each experiment, cells were exposed to the indicated concentrations of either compound under defined experimental conditions. Control groups were exposed to an equivalent volume of DMSO, adjusted to a final concentration of 0.5%, a level considered non-toxic in cell culture. This ensured that any observed effects could be attributed to the test compounds rather than the solvent.

### 2.7. Cell Viability Assay

Cell viability was assessed using the MTT assay. Briefly, cells were seeded at a density of 7000 cells per well in 96-well plates and allowed to adhere for 24 h. Cells were then treated under the indicated experimental conditions for the specified time points. MTT solution was added to each well at a final concentration of 0.5 mg/mL, followed by incubation for 3 h at 37 °C. The resulting formazan crystals were solubilized in 100 µL dimethyl sulfoxide (DMSO), and absorbance was measured at 570 nm using a CLARIOStar microplate reader. Cell viability was expressed as a percentage relative to the untreated vehicle control (0.5% DMSO).

### 2.8. Colony Formation Assay

For clonogenic survival analysis, cells were first seeded in 24-well plates at a density of 20,000 cells per well and allowed to attach for 24 h. After the indicated treatments, cells were harvested by trypsinization and re-plated into 6-well plates at 1000 cells per well in complete growth medium without TDB. Colonies were allowed to form over 7–14 days. At the end of the incubation period, cells were washed with PBS, fixed in 4% paraformaldehyde for 20 min, and stained with 2% crystal violet. Colonies containing ≥50 cells were counted manually, and clonogenic survival was expressed relative to the untreated vehicle control.

### 2.9. Flow Cytometry Analysis

Apoptotic and necrotic cell populations were assessed by Annexin V-FITC/propidium iodide (PI) staining followed by flow cytometric analysis. After treatment, cells were harvested by gentle trypsinization, washed twice with cold PBS, and resuspended in binding buffer. Annexin V-FITC and PI were added according to the manufacturer’s instructions (Immuno Tools, Friesoythe, Germany), and the cell suspension was incubated in the dark for 30 min at room temperature. Samples were analyzed using a Guava easyCyte HT flow cytometer (Merck KGaA, Darmstadt, Germany) with GuavaSoft Software (version 3.3). Data were expressed as the percentage of viable, early apoptotic, late apoptotic, and necrotic cells.

### 2.10. Scratch-Wound Assay

Cells were seeded at 5 × 10^4^ cells per well in 96-well plates and cultured until a confluent monolayer was formed. A uniform wound was created in each well using a sterile 200 µL pipette tip. Detached cells were gently removed by washing with PBS, and the monolayers were then treated with non-cytotoxic concentrations of TDB (6.25, 12.5, and 25 µM) in DMEM containing 1% FBS. Images were captured at 0, 3, 6, 12, and 24 h using a Nikon Eclipse Ts2 inverted microscope (10× magnification; China). Wound closure was quantified from the captured images with ImageJ software version 1.54p (NIH, Bethesda, MD, USA). Cell migration was expressed as the percentage of wound closure using the following formula:(1)$$\%Migration=\;\frac{A_{t0}-A_{th}}{A_{t0}}\;\times \;100$$
where *A*_*t*0_ is the wound area at 0 h and *A_th_* is the wound area at the indicated time point.

### 2.11. Western Blot Analysis

Cells were lysed in RIPA buffer supplemented with protease and phosphatase inhibitor cocktails to preserve protein integrity. Total protein content was determined using the Pierce™ BCA Protein Assay Kit (Thermo Fisher Scientific, Waltham, MA, USA). Equal amounts of protein were resolved by SDS–PAGE and transferred to PVDF membranes *via* a wet transfer system. Membranes were blocked in BlockPRO^TM^ 1 Min Protein-Free Blocking Buffer (Visual Protein, Taipei, Taiwan), followed by overnight incubation at 4 °C with specific primary antibodies. After three washes in TBS-T, membranes were incubated with horseradish peroxidase–conjugated secondary antibodies (1:2000) for 1 h at room temperature. Protein bands were visualized using Immobilon^®^ Western Chemiluminescent HRP Substrate (EMD Millipore Corporation, Burlington, MA, USA) and detected on an X-ray film developer. Band intensities were quantified by densitometric analysis using ImageJ software. For each sample, target protein signals were normalized to the corresponding GAPDH band from the same lane to correct for any loading or transfer variability. These normalized values, obtained from three independent biological replicates, were then used for statistical analyses. The unedited Western blot results are demonstrated in Appendix A and detailed information on primary and secondary antibodies (including working concentrations, catalog numbers, and suppliers) is listed in Appendix A.

### 2.12. In Silico Prediction of Blood–Brain Barrier Permeability

The test compound, 4,5,4′-trihydroxy-3,3′-dimethoxybibenzyl (TDB), was evaluated for its potential to cross the blood–brain barrier (BBB). Its Simplified Molecular Input Line Entry System (SMILES) notation—COC1CC(CCC2CC(O)C(O)C(OC)C2)CCC1O—was generated from the chemical structure and used as input for computational tools requiring this format. For platforms accepting chemical names, the full designation was provided.

BBB permeability was assessed using four publicly accessible web-based prediction models, each employing distinct algorithms, descriptor sets, and decision rules, thereby offering complementary perspectives:ADMETlab 3.0 (https://admetlab3.scbdd.com accessed on 29 July 2025):

A multi-endpoint ADMET prediction platform trained on over 400,000 curated compounds. BBB penetration is predicted *via* a multi-task Directed Message Passing Neural Network (DMPNN), providing both a probability score and categorical classification (BBB+ = permeable, BBB– = non-permeable). Classification is based on internal probability thresholds [20].

LogBB_Pred (http://ssbio.cau.ac.kr/software/logbb_pred/ accessed on 29 July 2025):

A regression-based model predicting the logarithmic brain-to-blood concentration ratio (logBB). Compounds with logBB ≥ –1 are classified as permeable, while values < –1 are classified as non-permeable [21].

LightBBB (http://ssbio.cau.ac.kr/software/bbb/ accessed on 29 July 2025):

A Light Gradient Boosting Machine (LightGBM)-based classifier trained on CNS and non-CNS drugs. Outputs are binary (permeable/non-permeable) based on learned probability thresholds, with reported classification accuracy of ~90% for CNS compounds [22]

BBB Predictor (Tree2C) (https://www.ddl.unimi.it/vegaol/bbbp.htm accessed on 29 July 2025):

A decision tree classifier that categorizes compounds as “permeant” or “not permeant” based on cheminformatics descriptors. The model requires the molecule in neutral form and is optimized for structural classes including bibenzyl derivatives, making it relevant for TDB assessment [23].

### 2.13. Statistical Analysis

Data are presented as mean ± standard error of the mean (SEM) from a minimum of three independent biological replicates. Statistical analyses were performed using GraphPad Prism (ver. 10.5.0 (673), GraphPad Software, Boston, MA, USA). Group comparisons were conducted using one-way ANOVA, and differences were considered statistically significant at *p* < 0.05. Exact *p*-values corresponding to each bar graph are provided in Appendix A for transparency.

## 3. Results

### 3.1. 4,5,4′-Trihydroxy-3,3′-dimethoxybibenzyl is a Potent Antioxidant Isolated from Dendrobium pachyglossum

Our prior phytochemical investigation of *Dendrobium pachyglossum* Par. & Rchb.f. (Figure 1A) led to the identification of several bibenzyl derivatives [12]. Among these, 4,5,4′-trihydroxy-3,3′-dimethoxybibenzyl (TDB; Figure 1B) was selected for further investigation due to its distinctive structural characteristics. TDB carries a bibenzyl backbone with three hydroxyl groups positioned in ortho and para orientations, which are commonly associated with hydrogen-donating and radical-scavenging activity [24]. In addition, two methoxy substituents are likely to enhance lipophilicity and stabilize radical intermediates, properties often linked to improved membrane interaction and antioxidant potential [25,26]. To experimentally validate these structural predictions, we assessed the antioxidant activity of TDB using two complementary assays. TDB exhibited appreciable activity, with a ferric reducing antioxidant power (FRAP) of 4056.08 ± 190.67 µM Trolox equivalents (TE)/mg and a DPPH radical scavenging capacity of 1974.62 ± 72.98 TE µM/mg. These findings indicate that TDB possesses appreciable antioxidant properties, supporting its relevance for further biological evaluation.

Plant-derived antioxidants have frequently been linked to anticancer effects through several mechanisms such as induction of apoptosis, modulation of oncogenic signaling pathways, and enhancement of chemotherapeutic efficacy [27,28,29]. Guided by this rationale, we next examined whether TDB exerts antitumor effects in glioblastoma multiforme (GBM), an aggressive brain tumor with limited treatment options. For this purpose, we employed the U87MG human glioblastoma cell line as a well-established *in vitro* model to assess the therapeutic potential of TDB.

### 3.2. TDB Reduces Cell Viability and Clonogenic Survival in U87MG Glioblastoma Cells

To investigate the cytotoxic potential of TDB against GBM, U87MG cells were treated with increasing concentrations of TDB (ranging from 6.25 to 200 µM) for 24, 48, and 72 h. Cell viability was assessed using the MTT assay. As shown in Figure 2A, TDB reduced cell viability in a dose- and time-dependent manner. After 24 h of treatment, only modest cytotoxicity was observed. Even at the highest concentration (200 µM), cell viability remained at approximately 74%, with no significant reduction seen at 50 µM or lower concentrations compared to untreated controls. At 48 h, TDB began to exert more pronounced effects. Cell viability declined to approximately 91%, 82%, 79%, and 46% at 25, 50, 100, and 200 µM, respectively, indicating a clear dose-dependent cytotoxic response. Prolonged exposure to TDB for 72 h further amplified these effects. Cell viability decreased to 77%, 69%, 59%, and 48% at 25, 50, 100, and 200 µM, respectively. Notably, the cytotoxic effect of 200 µM TDB at 72 h was comparable to that at 48 h, suggesting a potential saturation point in cell killing at high concentrations. Based on the enhanced and consistent cytotoxicity observed at the 72 h time point, this treatment window was selected for subsequent mechanistic and functional assays of this study.

To further assess the long-term impact of TDB on GBM cell survival, we conducted a clonogenic assay to examine the ability of U87MG cells to form colonies after treatment. In line with the MTT results, a 72 h exposure to TDB significantly suppressed colony formation in a concentration-dependent manner. Treatment with 25, 50, 100, and 200 µM TDB reduced clonogenicity to approximately 69%, 52%, 26%, and 7%, respectively, compared to untreated controls (Figure 2B,C). These results suggest that TDB not only reduces short-term cell viability but also compromises the long-term proliferative capacity of GBM cells. Together, these findings underscore the anti-proliferative potential of TDB and its capacity to hinder tumor cell renewal.

### 3.3. TDB Induces Apoptosis Cell Death in U87MG Glioblastoma Cells

Given the observed reduction in both cell viability and clonogenic potential, we further characterize the underlying mode of cell death triggered by TDB in U87MG cells. To distinguish between apoptotic and necrotic responses, we conducted flow cytometric analysis using Annexin V and propidium iodide (PI) staining following 72 h treatments. As shown in Figure 3A,B, TDB treatment at 50, 100, and 200 µM resulted in a substantial and concentration-dependent increase in apoptotic cell populations, reaching approximately 33%, 43%, and 47%, respectively. In contrast, treatment with 25 µM TDB produced apoptotic levels similar to the untreated control (approximately 20%). Notably, the proportion of necrotic cells remained relatively constant across all treatments, even at the highest concentration. These results indicate that TDB primarily induces apoptosis rather than necrosis, highlighting apoptotic cell death as a key contributor to its anti-GBM activity. A detailed distribution of cell populations across early apoptosis, late apoptosis, and necrosis is provided in Appendix A.

To further investigate the apoptotic response induced by TDB, we selected 100 µM as the experimental concentration for mechanistic analysis, based on our flow cytometry results showing a nearly two-fold increase in apoptotic cells compared to the untreated control. To validate the activation of apoptotic pathways at the molecular level, we performed a time-course study with Western blot analysis to assess the expression of key apoptotic regulators, including the pro-apoptotic protein Bax and the anti-apoptotic proteins Bcl-xL and Mcl-1. As shown in Figure 3C,D, Bax expression was upregulated as early as 3 h after TDB treatment and continued to increase over the course of the treatment, suggesting progressive activation of the apoptotic pathway. In contrast, Bcl-xL levels began to decline at 3 h post-treatment and steadily decreased over time, indicating a loss of survival signaling. Notably, Mcl-1 expression was gradually diminished and became nearly undetectable by 72 h, reflecting sustained apoptotic pressure. This reciprocal regulation of pro- and anti-apoptotic proteins highlights a shift in the intracellular balance toward cell death. Taken together, these findings further support apoptosis as a major mechanism underlying TDB-induced cytotoxicity in U87MG cells and are consistent with the Annexin V/PI flow cytometry results.

### 3.4. TDB Suppresses mTORC1/mTORC2 Activity in U87MG Glioblastoma Cells

The mammalian target of rapamycin (mTOR) signaling pathway plays a pivotal role in regulating cell growth, proliferation, metabolism, and survival. In GBM, hyperactivation of this pathway is frequently observed and is associated with increased tumor aggressiveness and resistance to therapy. mTOR mediates its effects through two distinct complexes: mTOR complex 1 (mTORC1), which promotes protein synthesis *via* ribosomal S6 kinase (S6K) and its downstream target S6, and mTOR complex 2 (mTORC2), which supports cell survival and metabolism through phosphorylation of Akt at Ser473 [30]. In this study, we focused on the phosphorylation status of S6 and Akt as representative markers of mTORC1 and mTORC2 activity, respectively. Western blot analysis revealed that TDB treatment resulted in a significant reduction in the levels of phosphorylated S6 (pS235/236-S6) and phosphorylated Akt (pS473-Akt) over a 72 h period (Figure 4), suggesting that TDB may attenuate the activity of both mTOR complexes. Notably, total protein levels of S6 and Akt were unaffected, indicating that TDB specifically modulates the activation state of these targets rather than altering their overall expression. These findings suggest that the anti-GBM activity of TDB may involve suppression of mTOR signaling, thereby interfering with key processes that promote tumor cell growth and survival.

### 3.5. TDB Attenuates Cell Migration and Mesenchymal Transition in U87MG Glioblastoma Cells

GBM is characterized not only by uncontrolled proliferation but also by its highly migratory nature, which plays a critical role in tumor invasiveness, recurrence, and resistance to conventional therapy [31,32]. To assess the potential impact of TDB on GBM cell migration, we conducted a scratch-wound healing assay using non-cytotoxic concentrations of TDB (≤25 µM), as established from our earlier MTT viability data (Figure 2A). As shown in Figure 5A,B, TDB treatment significantly delayed wound closure compared to the untreated control. Quantitative analysis revealed that cells exposed to TDB maintained a wider wound area throughout the observation period, reflecting impaired migratory activity. Notably, at the highest tested concentration (25 µM), the wound gap remained largely unclosed even at 24 h, whereas control cells had fully closed the wound. These results indicate that, beyond its cytotoxic effects, TDB can effectively suppress the migratory behavior of GBM cells.

Epithelial–mesenchymal transition (EMT) is a dynamic cellular reprogramming process in which cells lose epithelial characteristics and acquire mesenchymal properties, enhancing their motility and invasive capacity [33]. In GBM, EMT-like processes contribute to tumor cell migration, invasion into surrounding brain tissue, and resistance to therapy. A hallmark of EMT is the upregulation of mesenchymal markers such as N-cadherin, which promotes cell detachment and tissue invasion. This phenotypic shift is orchestrated by a set of transcription factors including Snail, Slug, Twist1, and ZEB1, which repress expression of epithelial markers and activate genes associated with cytoskeletal remodeling and motility [17,34,35]. To determine whether TDB affects the EMT phenotype in GBM cells, we examined the expression levels of key EMT markers following TDB exposure. Western blot analysis revealed that treatment with TDB for 6 h led to a reduction in N-cadherin expression, along with downregulation of the transcription factors Snail, Slug, Twist1, and ZEB1 (Figure 5C,D). These changes were detectable at all tested concentrations (6.25–25 µM), with the most pronounced effect observed at 25 µM. These findings suggest that TDB interferes with the transcriptional regulation that underlies the mesenchymal phenotype in GBM cells. Together with results from the scratch-wound assay, our data indicate that TDB impairs GBM cell migration at least in part through inhibition of EMT. This suppression of EMT may represent a mechanistic basis for the anti-migratory effect of TDB in GBM.

### 3.6. TDB Improves the Temozolomide Efficacy Against GBM Cells

Temozolomide (TMZ) remains the standard chemotherapeutic agent for GBM; however, its clinical efficacy is frequently limited by the development of resistance [36]. Therefore, identifying agents that can enhance TMZ’s anti-cancer activity is of critical importance for improving therapeutic outcomes. To evaluate the baseline response of GBM cells to TMZ, we performed a clonogenic survival assay in U87MG cells treated with increasing concentrations of TMZ (2.5–50 µM) for 72 h. TMZ treatment at 2.5, 5, and 10 µM reduced colony formation by 19%, 24%, and 26%, respectively, while complete suppression of clonogenicity was observed at 25 and 50 µM (Figure 6A). Based on these results, 10 µM TMZ was selected for combination studies, as it produced a partial inhibitory effect, allowing us to examine whether TDB could further enhance TMZ cytotoxicity.

We next investigated whether TDB could enhance the response of GBM cells to TMZ. In U87MG cells, co-treatment with TMZ (10 µM) and TDB (25 or 50 µM) reduced colony formation by 49% and 53%, respectively, representing roughly a twofold improvement compared with TMZ alone (Figure 6B,C). To extend these findings into a more clinically relevant setting, we evaluated the combination in Jx22, a patient-derived xenograft (PDX) GBM cell line. PDX models preserve critical features of primary tumors, including molecular heterogeneity and therapeutic resistance, thereby offering higher translational relevance [19]. In Jx22 cells, TMZ alone (10 µM) reduced clonogenic survival by 28%, whereas the addition of TDB (25 or 50 µM) achieved an almost complete suppression of colony formation (~96% reduction; Figure 6B,D). This corresponds to an approximately 3.4-fold enhancement over TMZ monotherapy, underscoring the remarkable potentiation of TMZ’s effect by TDB in a patient-derived context. Collectively, these results highlight TDB as a promising adjunctive candidate that could help overcome resistance and improve the therapeutic efficacy of TMZ in GBM.

### 3.7. TDB Shows Potential to Permeate the Blood–Brain Barrier

The blood–brain barrier (BBB) represents a major pharmacological challenge for developing therapeutics aimed at treating brain malignancies such as GBM, as it restricts the entry of most small molecules into the central nervous system [37]. To explore whether TDB possesses the capacity to traverse the BBB and reach intracranial targets, we conducted a series of *in silico* predictions using four independent computational platforms: ADMETlab 3.0, LogBB_Pred, LightBBB, and BBB Predictor (Tree2C). These models employ distinct prediction strategies, including chemical structure–based classification rules, quantitative structure–activity relationship (QSAR) algorithms, and deep learning frameworks, thereby providing complementary perspectives on the likelihood of BBB penetration [20,21,22,23].

When analyzed by ADMETlab 3.0 using the SMILES notation of TDB, the model yielded a BBB penetration probability of 0.022 and classified the compound as “Category 1: BBB+,” suggesting a potential to permeate the BBB (Appendix A). The LogBB_Pred model, which estimates the logarithmic brain-to-blood concentration ratio, calculated a LogBB value of −0.42844 for TDB. As compounds with LogBB ≥ −1 are considered permeable, TDB was assigned the classification “BBB Permeable” (Appendix A). Similarly, the LightBBB model predicted that TDB is “Permeable” based on a deep learning–driven categorical assessment. Finally, the BBB Predictor (Tree2C) model, using the compound’s chemical name, returned a binary output of “BBB permeant: Yes,” indicating a positive prediction for BBB penetration (Appendix A).

Collectively, all four computational approaches consistently predicted that TDB is capable of crossing the BBB (Table 1), supporting the potential for TDB to exert its pharmacological activity within the brain and providing a rationale for further *in vivo* pharmacokinetic and efficacy studies.

## 4. Discussions

Natural products from plants have long been a cornerstone of drug discovery, offering a rich diversity of phytochemicals with distinctive structural complexity and broad pharmacological potential. Among these, orchids of the genus *Dendrobium* have attracted particular attention for their abundance of bioactive constituents exhibiting anti-inflammatory [13,38], antioxidant [12,13,38], and anticancer properties [5,6,7,8,9,10,11]. Bibenzyl derivatives are among the major active compounds in *Dendrobium*, and many have demonstrated promising anticancer activities across various tumor models [5,6,7,8,9,10,11]. *Dendrobium pachyglossum* Par. & Rchb.f., in particular, contains several unique bibenzyl and phenanthrene derivatives with notable pharmacological profiles [12]. In this study, we investigated the anti-glioblastoma potential of 4,5,4′-trihydroxy-3,3′-dimethoxybibenzyl (TDB), a bibenzyl derivative isolated from *D. pachyglossum*, and characterized its multiple tumor-suppressive activities in glioblastoma multiforme (GBM), a highly aggressive and treatment-refractory brain tumor.

Our findings demonstrate that TDB exerts potent anti-GBM activity through multiple mechanisms, including inhibition of proliferation, induction of apoptosis, suppression of mTOR signaling, attenuation of epithelial–mesenchymal transition (EMT), and enhancement of temozolomide (TMZ) efficacy. In U87MG glioblastoma cells, TDB reduced cell viability in a dose- and time-dependent fashion (Figure 2A). Notably, its sustained suppression of clonogenic growth (Figure 2B,C) indicates that TDB interferes with the self-renewal capacity of GBM cells, a property closely associated with tumor recurrence and therapeutic resistance. These inhibitory effects were not confined to U87MG cells. In H4 neuroglioma cells, representing a low-grade glioma model [16], TDB similarly reduced cell viability and impaired clonogenic potential (Appendix A). Importantly, activity was also observed in patient-derived Jx22 GBM cells, where TDB decreased viability in a dose-dependent manner (Appendix A), reinforcing its activity in clinically relevant contexts. Together, these findings highlight the consistent ability of TDB to compromise both immediate survival and long-term proliferative potential across glioma models, underscoring its promise as a candidate therapy for GBM.

Mechanistic analyses revealed that TDB primarily induces apoptotic cell death in GBM cells. In U87MG cells, flow cytometric analysis demonstrated a concentration-dependent increase in apoptotic populations, while necrotic cell death remained largely unchanged (Figure 3A,B and Appendix A). Interestingly, H4 neuroglioma cells exhibited an even greater magnitude of apoptosis in response to TDB, with a marked dose-dependent increase in apoptotic fractions (Appendix A). This observation suggests that TDB’s pro-apoptotic effects are not only consistent across glioma models but may be particularly pronounced in low-grade glioma, underscoring its potential to target tumor cells of varying aggressiveness. Supporting findings on induction of apoptosis, immunoblotting in U87MG showed time-dependent upregulation of the pro-apoptotic protein Bax, accompanied by downregulation of the pro-survival proteins Bcl-xL and Mcl-1 (Figure 3C,D). This reciprocal regulation of apoptotic mediators reflects a decisive shift in the intracellular balance toward cell death. The marked reduction in Mcl-1—an anti-apoptotic factor frequently associated with chemoresistance and poor prognosis in cancer patients [39]—suggests that TDB may also sensitize GBM cells to other therapeutic agents, further enhancing its potential as an effective treatment strategy.

In addition to triggering apoptosis, TDB suppressed both mTORC1 and mTORC2 activity, as evidenced by decreased phosphorylation of S6 and Akt, respectively, without altering their total protein levels (Figure 4). The mTOR pathway is frequently hyperactivated in GBM and is strongly linked to enhanced tumor cell survival, metabolic adaptation, and aggressive growth [40,41]. By concurrently targeting both mTOR complexes, TDB may interfere with distinct yet complementary signaling branches—mTORC1-driven protein synthesis and mTORC2-mediated survival and cytoskeletal regulation. Such dual inhibition could disrupt critical oncogenic processes that sustain glioblastoma progression, contributing substantially to the anti-proliferative effects observed in our study.

While our data highlight apoptosis induction and suppression of mTORC1/2 signaling as key downstream effects of TDB, the precise upstream molecular targets remain to be defined. Although the precise upstream molecular targets of TDB remain undefined, studies on structurally related bibenzyl derivatives offer valuable mechanistic insights regarding the upstream events that may drive TDB-induced apoptosis. Notably, other naturally occurring bibenzyl compounds such as gigantol, erianin, and moscatilin have been extensively studied for their anticancer properties and provide useful parallels for interpreting the potential mechanisms of TDB.

Gigantol has been shown to disrupt PI3K/Akt/mTOR signaling in breast cancer cells [42] and to inhibit the PI3K/Akt/NF-κB pathway in hepatic cancer, leading to apoptotic cell death [43]. In cervical cancer, gigantol triggered apoptosis through ROS generation, lipid peroxidation, oxidative stress, and attenuation of Wnt/β-catenin signaling [44]. Similarly, erianin exerts pro-oxidant effects by inducing ROS accumulation, glutathione depletion, and lipid peroxidation, culminating in ferroptosis in lung cancer cells [5]. In other contexts, erianin activates ROS-mediated apoptosis and inhibits JAK/STAT signaling in esophageal carcinoma [45], while in pancreatic cancer it modulates the AKT/FOXO1 and ASK1/JNK/p38 MAPK pathways to suppress proliferation and migration [8]. Moreover, erianin-induced apoptosis and autophagy in osteosarcoma were linked to ROS/JNK signaling cascades [46]. Another bibenzyl derivative, moscatilin, has been reported to activate JNK signaling and trigger DNA damage in colorectal cancer cells [47], while promoting apoptosis *via* JNK activation in head and neck carcinoma and pancreatic cancer models [48,49]. These findings collectively suggest that bibenzyl compounds can modulate multiple upstream regulators, particularly PI3K/Akt, ROS-mediated oxidative distress, and JNK cascades. Based on these precedents, it is plausible that TDB exerts its pro-apoptotic effects through similar upstream mechanisms, which ultimately converge on mTOR suppression and apoptotic signaling observed in our study. Future investigations will be required to delineate these upstream events and clarify the direct molecular targets of TDB in GBM.

TDB also impaired the migratory capacity of GBM cells. At non-cytotoxic concentrations, TDB significantly delayed wound closure in a scratch assay and downregulated the expression of key mesenchymal markers and EMT-associated transcription factors, including N-cadherin, Snail, Slug, Twist1, and ZEB1 (Figure 5). Given that EMT is a central driver of GBM invasiveness and therapeutic resistance [17], its suppression by TDB adds another mechanistic dimension to its anti-tumor profile. Notably, similar EMT-inhibitory effects have been observed in non-small cell lung cancer cells treated with TDB derived from a different *Dendrobium* species, where the compound also demonstrated selective cytotoxicity toward malignant cells while sparing normal counterparts [7]. This convergence of evidence suggests that TDB’s anti-migratory and potentially cancer-selective actions may be conserved across cancer types. Beyond TDB, other bibenzyl derivatives such as erianin [9], moscatilin [50], and gigantol [51] have likewise been shown to suppress EMT in diverse malignant cancer models, underscoring the multifaceted roles of this compound class in regulating both tumor survival and metastatic progression. Taken together, these findings suggest that the anti-migratory activity of TDB may represent a conserved property across bibenzyl scaffolds and cancer types, thereby strengthening the case for their therapeutic relevance.

Given the limited success of current GBM treatments and the persistent challenge of TMZ resistance [36], identifying agents that can enhance TMZ efficacy is of high clinical relevance. In our study, TDB markedly improved the cytotoxic effect of TMZ in both U87MG cells and the patient-derived xenograft line Jx22. Co-treatment significantly suppressed clonogenic survival compared with TMZ alone, with an approximately twofold enhancement in U87MG cells and a more pronounced ~3.4-fold enhancement in Jx22 cells. (Figure 6B–D). These findings suggest that TDB may help overcome both inherent and acquired TMZ resistance—one of the major barriers to durable therapeutic responses in GBM [52]. The chemosensitizing effect observed here highlights TDB’s potential to improve treatment outcomes when incorporated into combination regimens, providing a strong rationale for its further evaluation in preclinical and clinical settings.

Because the therapeutic success of TDB depends on its ability to reach tumor sites within the brain, we next assessed its blood–brain barrier (BBB) permeability using four independent *in silico* models: ADMETlab 3.0, LogBB_Pred, LightBBB, and BBB Predictor (Tree2C). All platforms consistently classified TDB as BBB-permeable (Table 1). This prediction is supported by its structural attributes: the bibenzyl scaffold, decorated with hydroxyl and methoxy groups, offers a favorable balance between lipophilicity and polarity, enabling passive diffusion across the lipid-rich BBB endothelium. Its molecular weight is well below the ~400 Da threshold for CNS penetration, and the lipophilic influence of methoxy groups counteracts the potential hydrogen-bonding limitation of hydroxyl groups [53]. Taken together, these computational assessments suggest that TDB may possess physicochemical features favorable for CNS delivery, although experimental validation will be necessary to confirm its BBB permeability and translational potential for GBM therapy.

It is important to acknowledge several limitations of the present study. Although our work provides compelling *in vitro* evidence of TDB’s anti-GBM activity, further *in vivo* validation, particularly in patient-derived orthotopic xenograft models, is required to establish its therapeutic relevance under physiologically representative conditions [54]. Orthotopic implantation of patient-derived GBM cells into the mouse brain more faithfully mirrors the clinical scenario, recapitulating the complex tumor microenvironment, including diffuse infiltration along white matter tracts, hypoxia-induced heterogeneity, and interactions with stromal and immune elements [55,56,57,58]. Unlike traditional ectopic xenografts or two-dimensional culture systems, these models also enable evaluation of clinically meaningful endpoints, such as intracranial tumor progression, neurological impairment, and overall survival [59,60,61].

Another limitation involves the prediction of BBB permeability, which in this study relied solely on *in silico* modeling. While computational approaches are valuable as an initial screen, they cannot fully capture the complexities of BBB transport, including active influx/efflux systems, endothelial tight junctions, and local metabolic activity. Consequently, experimental validation is essential. Future studies will therefore employ complementary strategies such as *in vitro* Parallel Artificial Membrane Permeability (PAMPA)-BBB assays [62], 3D multicellular BBB spheroids [63], and ultimately *in vivo* pharmacokinetic or imaging analyses to determine whether TDB can effectively penetrate the BBB and accumulate in brain tumors.

A further consideration concerns the lack of non-cancerous astrocytes or brain-derived cells in our experimental design to directly assess selective cytotoxicity. While this remains an important next step for evaluating translational feasibility, prior works in non-small cell lung cancer (NSCLC) provide supportive evidence for cancer cell specificity [7,11]. In those studies, TDB exhibited markedly selective anticancer activity, with IC_50_ values of approximately 100–200 µM in NSCLC cells, whereas the IC_50_ in normal human dermal papilla cells exceeded 300 µM. Notably, the maximum concentration used in our GBM experiments (200 µM) falls below the threshold at which cytotoxicity was observed in non-cancerous cells, indirectly supporting the presence of a therapeutic window. Nevertheless, future studies should validate these findings in normal astrocytes or other non-cancerous brain cells to more rigorously confirm GBM selectivity.

An additional limitation relates to the absence of formal quantification of drug–drug interactions between TDB and TMZ using methods such as the Chou-Talalay Combination Index (CI). Although we attempted such analyses with both the MTT assay and the clonogenic assay, technical challenges prevented us from generating reliable data across the dynamic range required for CI calculation. In MTT assays, the relative resistance of U87MG cells combined with the solubility constraints of both compounds restricted the concentration ranges that could be tested. In contrast, in clonogenic assays, the steep dose–response profile of TMZ posed a different challenge, as 10 µM reduced colony formation to ~70% while 25 µM completely abolished it. This narrow response range made it very challenging to estimate a precise IC50 suitable for combination analysis. For these reasons, we presented clonogenic survival results only to demonstrate the combinatorial impact of TDB and TMZ, while deliberately avoiding definitive claims of “synergistic” or “additive” effects. Future studies employing optimized assay conditions and systematic combination designs will be necessary to rigorously establish the nature of this interaction.

In addition, the precise molecular targets of TDB remain to be elucidated. Multi-omics approaches, encompassing proteomic, transcriptomic, and computational analyses, will be needed to clarify its mechanisms of action. Finally, systematic pharmacokinetic and toxicological evaluations in preclinical animal models will be crucial for defining a therapeutic window and safety profile. Taken together, these future investigations, particularly orthotopic xenograft validation, rigorous BBB permeability testing, TDB’s selectivity and comprehensive toxicological studies, represent indispensable next steps for establishing the translational potential of TDB as a candidate therapy for GBM.

## 5. Conclusions

This study identifies TDB from *D. pachyglossum* as a promising natural antioxidant with potent anti-glioblastoma activity. TDB exerts multifaceted effects, including suppression of proliferation, induction of apoptosis, inhibition of mTOR signaling, attenuation of EMT-driven migration, and enhancement of TMZ efficacy—even in a patient-derived GBM model. Notably, *in silico* BBB permeability analyses using four independent prediction models (ADMETlab 3.0, LogBB_Pred, LightBBB, and BBB Predictor) consistently suggested that TDB may be capable of crossing the BBB, with structural features supportive of passive diffusion into the CNS. This predictive permeability strengthens the rationale for further development, as effective brain delivery remains a critical barrier in GBM therapy. By targeting multiple hallmarks of GBM, demonstrating synergy with standard chemotherapy, and exhibiting physicochemical features potentially favorable for CNS delivery, TDB emerges as a compelling candidate for further preclinical and clinical advancement. With additional *in vivo* validation and comprehensive pharmacological characterization, TDB could contribute to the development of plant-derived agents capable of overcoming the formidable therapeutic challenges posed by GBM.

## Figures and Tables

**Figure 1 antioxidants-14-01212-f001:**
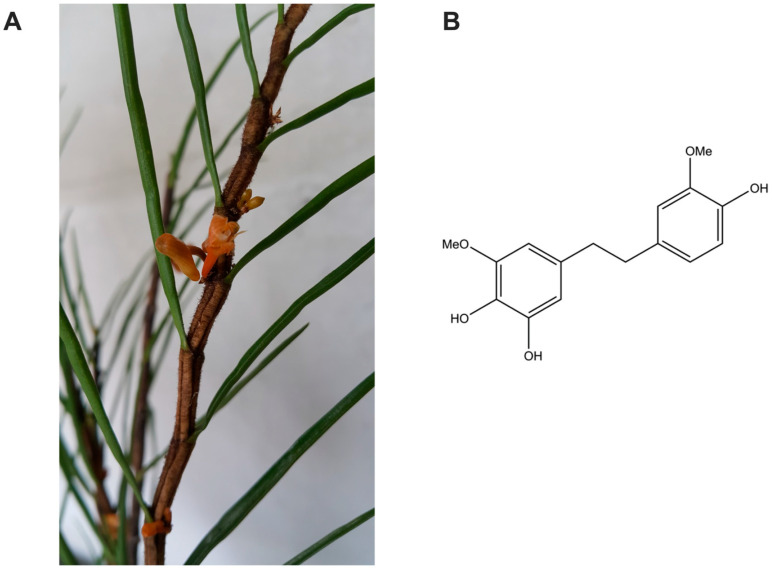
**Source and chemical structure of TDB.** (**A**) Photograph of *Dendrobium pachyglossum*, the orchid species from which the TDB was isolated. (**B**) Chemical structure of 4,5,4′-trihydroxy-3,3′-dimethoxybibenzyl (TDB).

**Figure 2 antioxidants-14-01212-f002:**
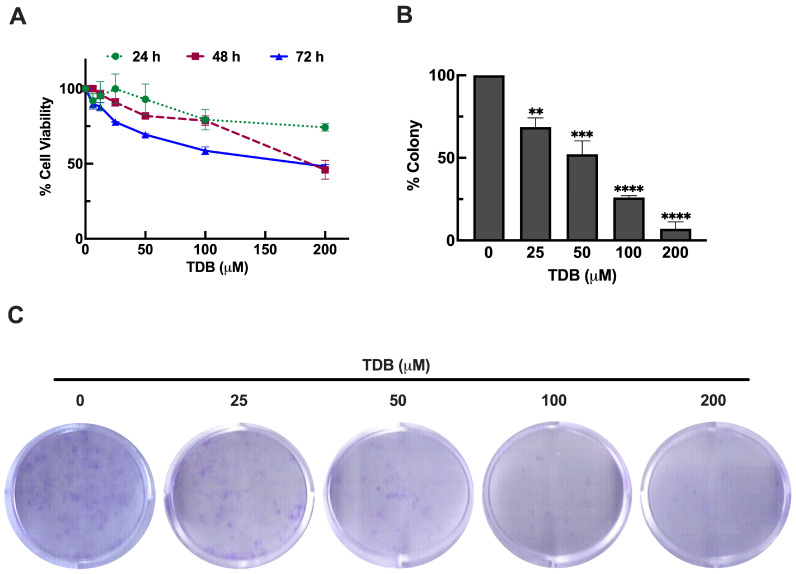
**TDB suppresses the proliferation and clonogenic survival of U87MG cells.** (**A**) Cells were treated with increasing concentrations of TDB (6.25–200 µM) for 24, 48, and 72 h, and cell viability was assessed using the MTT assay. (**B**,**C**) Clonogenic survival was evaluated after treatment with TDB (25, 50, 100, and 200 µM) for 72 h, followed by colony formation assessment. Data are presented as the mean ± SEM. MTT assay: *n* = 3 biological replicates, each with 4 technical replicates; clonogenic assay: n = 3 biological replicates, each with 3 technical replicates. ** *p* < 0.01, *** *p* < 0.001, **** *p* < 0.0001 vs. untreated controls.

**Figure 3 antioxidants-14-01212-f003:**
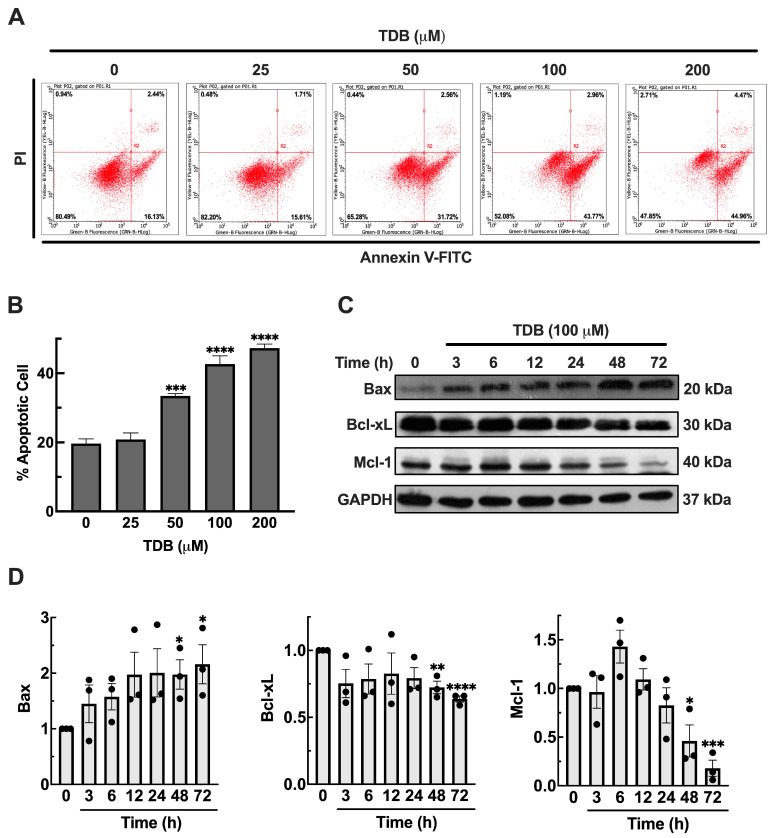
**TDB induces apoptosis in U87MG cells.** (**A**,**B**) Cells were treated with TDB (25, 50, 100, and 200 µM) for 72 h, and apoptosis was quantified by flow cytometry with Annexin V-FITC/PI dual staining. The apoptotic population included both early and late apoptotic cells. (**C**,**D**) Western blot analysis showed upregulation of pro-apoptotic Bax and downregulation of anti-apoptotic Bcl-xL and Mcl-1. Band intensities were normalized to GAPDH, and quantification is shown in (**D**). Data represent mean ± SEM from three independent biological replicates for both flow cytometry and Western blot analysis. Individual dots on bar graphs represent values from each biological replicate. * *p* < 0.05, ** *p* < 0.01, *** *p* < 0.001, **** *p* < 0.0001 vs. untreated controls.

**Figure 4 antioxidants-14-01212-f004:**
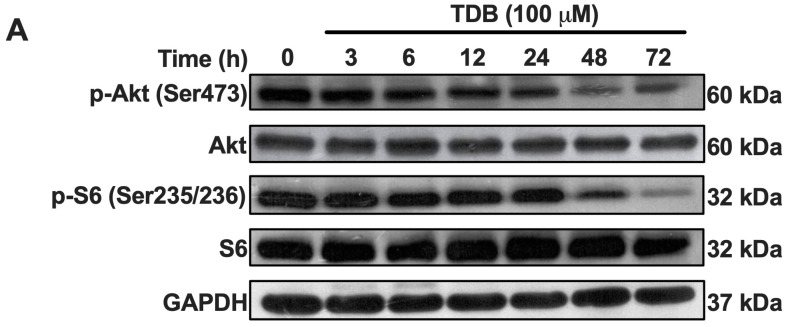
**TDB disrupts mTORC1 and mTORC2 signaling in U87MG cells.** (**A**,**B**) U87MG cells were treated with TDB (100 µM) for the indicated times. Phosphorylation of S6 (Ser235/236; downstream of mTORC1) and Akt (Ser473; downstream of mTORC2) was analyzed by Western blotting. Band intensities were normalized to GAPDH, with quantitation displayed in (**B**). Data are presented as the mean ± SEM from three independent biological replicates. Each dot on the bar graphs corresponds to an individual biological replicate. * *p* < 0.05, ** *p* < 0.01, *** *p* < 0.001 vs. untreated controls.

**Figure 5 antioxidants-14-01212-f005:**
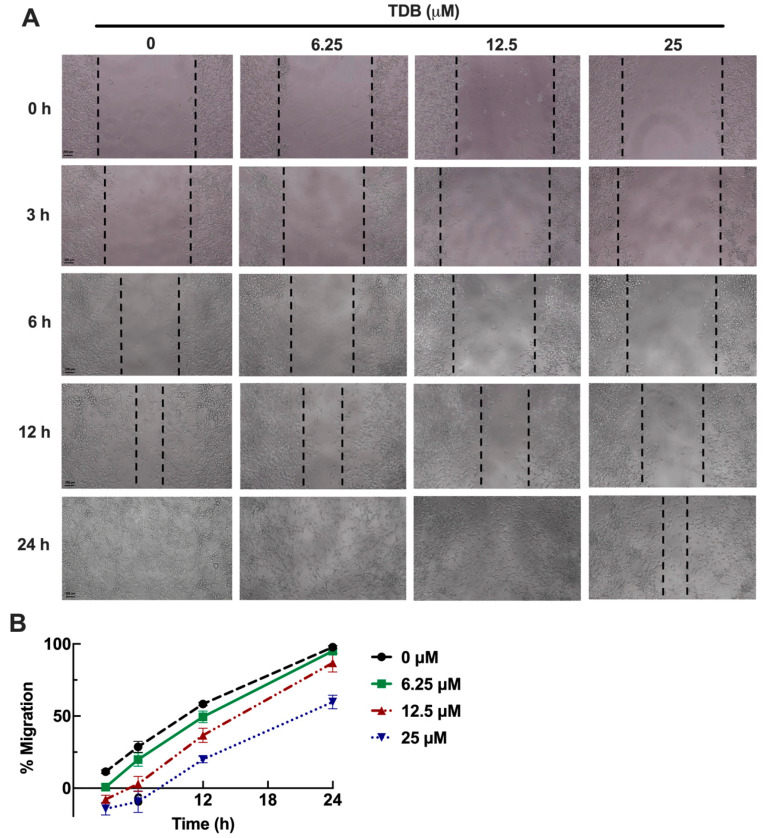
**TDB inhibits migration and suppresses epithelial–mesenchymal transition (EMT) markers in U87MG cells.** (**A**,**B**) U87MG cells were treated with TDB (6.25, 12.5, 25 µM) immediately after wound creation, and migration was monitored at multiple time points. Representative images were captured at 10× magnification (scale bar = 200 µm). Wound closure percentage was quantified and plotted in (**B**). *n* = 3 biological replicates, each with 3 technical replicates. Data are expressed as the mean ± SEM. (**C**,**D**) EMT-related proteins were evaluated by Western blotting after 6 h of TDB treatment. Band intensities were normalized to GAPDH and summarized in (**D**). Western blot data represent three independent biological replicates. Each dot on the bar graphs represents an individual biological replicate. Data are expressed as the mean ± SEM. * *p* < 0.05, ** *p* < 0.01, *** *p* < 0.001 vs. untreated controls.

**Figure 6 antioxidants-14-01212-f006:**
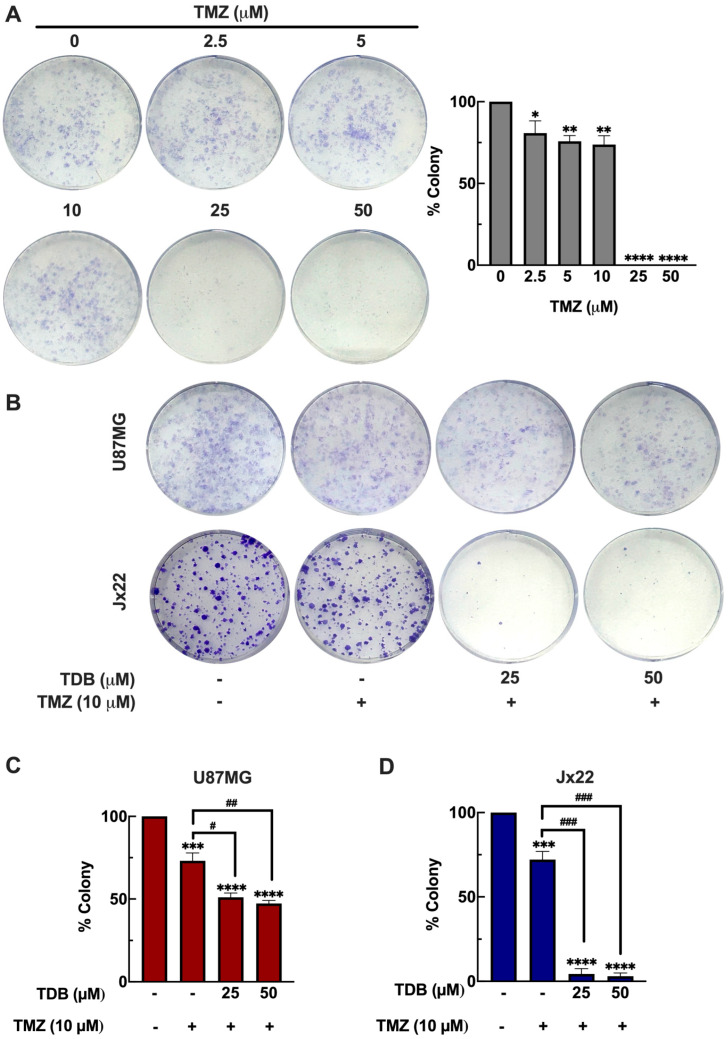
**TDB enhances the anticancer efficacy of temozolomide in U87MG and Jx22 patient-derived xenograft cells.** (**A**) U87MG cells were treated with various concentrations of temozolomide (TMZ; 2.5, 5, 10, 25, and 50 µM) for 72 h, and clonogenic survival was subsequently assessed. (**B**–**D**) U87MG and Jx22 cells were exposed to TMZ alone or in combination with TDB for 72 h, followed by a clonogenic survival assay. The representative colony formation images are shown in (**B**), while quantification of surviving colonies is presented in (**C**) (U87MG) and (**D**) (Jx22). Data are expressed as the mean ± SEM from three independent biological replicates. * *p* < 0.05, ** *p* < 0.01, *** *p* < 0.001, and **** *p* < 0.0001 vs. untreated controls; # *p* < 0.05, ## *p* < 0.01, and ### *p* < 0.001 vs. TMZ alone.

**Table 1 antioxidants-14-01212-t001:** Summary of *In Silico* Blood–Brain Barrier Permeability Predictions for TDB.

*In Silico* Model	Prediction Outcome	Quantitative Value/Category Rule
ADMETlab 3.0	BBB+ (Category 1)	Probability = 0.022
LogBB_Pred	BBB Permeable	LogBB value = −0.42844 (LogBB ≥ −1 for Permeable)
LightBBB	Permeable	Categorical prediction
BBB Predictor (Tree2C)	BBB permeant: Yes	Binary classification

## Data Availability

The original contributions presented in this study are included in the article/Appendix A. Further inquiries can be directed to the corresponding author.

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
