# Peer review of "A Bibenzyl from Dendrobium pachyglossum Exhibits Potent Anti-Cancer Activity Against Glioblastoma Multiforme"

_antioxidants, 2025, doi:10.3390/antiox14101212_

Round 1

Reviewer 1 Report

The manuscript by Aung et al. investigates the anti-glioblastoma potential of 4,5,4′-trihydroxy-3,3′-dimethoxybibenzyl (TDB), a bibenzyl derivative from Dendrobium pachyglossum. The study is well structured, provides mechanistic insights into apoptosis and mTOR signaling, and highlights synergistic effects with temozolomide. The inclusion of patient-derived xenograft (PDX) Jx22 cells increases the translational value. The manuscript is generally well written and fits the scope of Antioxidants. However, several issues need to be addressed before publication.

Major Comments

  1. The study relies entirely on in vitro and in silico approaches. While the PDX cell line strengthens translational potential, orthotopic xenograft or animal models would provide stronger support. The authors should discuss this limitation more explicitly in the Discussion and outline specific plans for in vivo validation.
  2. BBB permeability is predicted only through computational tools. Although the authors acknowledge this limitation, additional experimental evidence (e.g., PAMPA-BBB assay, in vitro endothelial monolayer permeability) would substantially increase credibility. At minimum, the Discussion should more critically evaluate the limitations of in silico predictions.
  3. The study highlights apoptosis induction (Bax upregulation, Bcl-xL/Mcl-1 downregulation) and mTORC1/2 suppression. However, the upstream molecular targets of TDB remain undefined. The authors should expand the Discussion to speculate on possible direct targets (e.g., PI3K/Akt, ROS modulation, mitochondrial stress), citing relevant literature on bibenzyl derivatives.
  4. No experiments were conducted in normal astrocytes or non-cancerous brain cells to evaluate selective cytotoxicity. Including such controls, or at least discussing this limitation, would be important for assessing translational feasibility.
  5. The claim that TDB enhances TMZ efficacy is promising. However, the study only presents colony counts without synergy quantification. The authors should calculate and present a combination index (CI; e.g., Chou–Talalay method) to clearly demonstrate synergism versus additivity.
  6. In several results (e.g., apoptosis assays, migration assays), the biological replicate numbers are low (n=3). Please clarify whether these are independent biological replicates or technical replicates, and provide exact p-values in Supplementary Data for transparency.
  7. The study focuses mainly on U87MG and one PDX-derived GBM cell line. While the data are compelling, using only a single tumor type limits the generalizability of the findings. At least three independent cancer cell lines (e.g., additional GBM models such as LN229, T98G, or even non-brain tumor lines) are usually recommended to demonstrate broader anticancer activity. The authors should either expand their dataset or explicitly discuss this limitation.
  1. Figures 2–6 would benefit from including representative images with scale bars (e.g., colony assays, wound healing). Current resolution in some panels is insufficient for evaluation. Western blot quantifications should clearly indicate normalization method and statistical testing.
  2. The Discussion could be strengthened by citing more recent work on bibenzyls in cancer (e.g., erianin’s role in ferroptosis and EMT suppression, 2020–2024 studies).
  3. Please specify solvent concentration controls (DMSO final concentration) across all experiments.
  4. Minor English corrections are required (e.g., “substantial proportion of modern anticancer drugs are either directly derived” → “have been directly derived”). A careful proofreading will improve readability.

Author Response

We have provided our detailed point-by-point responses in the attached pdf file for your kind consideration.

Reviewer 2 Report

This paper describes the bioactivity of the drug TDB against Glioblastoma Multiforme. Besides the quantification of the activity the authors made a detail analysis of the drug mechanisms against this cell line. Although the drug show activity against this cell line, a huge amount of experimental work is necessary to conclude about the possible application in Glioblastoma Multiforme treatment. For this reason the title of the paper should focus on the specific cell line that was used in this paper and remove Glioblastoma Multiforme from the title. Also, further experimental evidence must be included in this work about the cytotoxicity of the drug on non-cancer cells and compare the activity of the drug with a standard drug used in the treatment of this cancer. 

The results that are presented in this paper are well presented. As discussed previously the main problem of this paper is the missing information that must be presented to support the obtained conclusions. 

Author Response

(The authors gave the same response as above.)

Round 2

Reviewer 1 Report

Accept in present form.

Accept in present form.

Reviewer 2 Report

The authors have made some improvements in the original paper.

The authors have made some improvements in the original paper.